# FusionRCNN: LiDAR-Camera Fusion for Two-Stage 3D Object Detection

Xinli Xu [1], Shaocong Dong [1], Tingfa Xu [1,2], Lihe Ding [1], Jie Wang [1], Peng Jiang [3], Liqiang Song [3] and Jianan Li [1,*]

[1] Image Engineering & Video Technology Lab, School of Optics and Photonics, Beijing Institute of Technology, Beijing 100081, China
[2] Big Data and Artificial Intelligence Laboratory, Beijing Institute of Technology Chongqing Innovation Center(BITCQIC), Chongqing 401135, China
[3] National Astronomical Observatories of China, Beijing 100107, China
[*] Correspondence: lijianan@bit.edu.cn

**Abstract:** Accurate and reliable perception systems are essential for autonomous driving and robotics. To achieve this, 3D object detection with multi-sensors is necessary. Existing 3D detectors have significantly improved accuracy by adopting a two-stage paradigm that relies solely on LiDAR point clouds for 3D proposal refinement. However, the sparsity of point clouds, particularly for faraway points, makes it difficult for the LiDAR-only refinement module to recognize and locate objects accurately. To address this issue, we propose a novel multi-modality two-stage approach called **FusionRCNN**. This approach effectively and efficiently fuses point clouds and camera images in the Regions of Interest (RoI). The FusionRCNN adaptively integrates both sparse geometry information from LiDAR and dense texture information from the camera in a unified attention mechanism. Specifically, FusionRCNN first utilizes RoIPooling to obtain an image set with a unified size and gets the point set by sampling raw points within proposals in the RoI extraction step. Then, it leverages an intra-modality self-attention to enhance the domain-specific features, followed by a well-designed cross-attention to fuse the information from two modalities. FusionRCNN is fundamentally plug-and-play and supports different one-stage methods with almost no architectural changes. Extensive experiments on KITTI and Waymo benchmarks demonstrate that our method significantly boosts the performances of popular detectors. Remarkably, FusionRCNN improves the strong SECOND baseline by 6.14% mAP on Waymo and outperforms competing two-stage approaches.

**Keywords:** 3D object detection; LiDAR-camera fusion; two-stage

## 1. Introduction

Accurate 3D object detection is a crucial task in the fields of autonomous driving and robotics, where multiple sensors are utilized to capture comprehensive spatial information. Self-driving vehicles, for instance, commonly incorporate various sensors such as the IMU, radar, LiDAR, and camera. Among these sensors, LiDAR sensors possess a distinct advantage in obtaining precise depth and shape information, resulting in previous methods relying solely on point clouds achieving competitive performance. Additionally, some recent methods have substantially improved by incorporating a two-stage refinement module. These findings have inspired researchers to explore more effective LiDAR-based two-stage detectors further.

Two-stage 3D object detection methods can be classified into three primary categories based on the Point of Interest representation, namely, point-based, voxel-based, and point-voxel-based. Point-based approaches [1–4] utilize input sampling points to obtain point features for RoI refinement. Voxel-based techniques [5,6] rasterize point clouds into voxel-grids and extract features from 3D CNNs for refinement. Point-Voxel-based methods [7,8]

combine both feature learning schemes to enhance detection performance. However, regardless of the representation used, the sparse and non-uniform distribution characteristics of point clouds make it challenging to distinguish and locate objects at far distances, leading to false or missed detections, as demonstrated in Figure 1. These challenges are exacerbated when proposals contain only a few points (1–5), from which it is challenging to obtain enough semantic information. In urban scenarios, multi-sensor fusion performs better than single sensors in various tasks such as remote sensing [9,10]. Fortunately, cameras provide dense texture information and are complementary to LiDAR. Thus, designing a LiDAR-Camera fusion paradigm in two-stage detectors to leverage their complementary strengths effectively is of great importance.

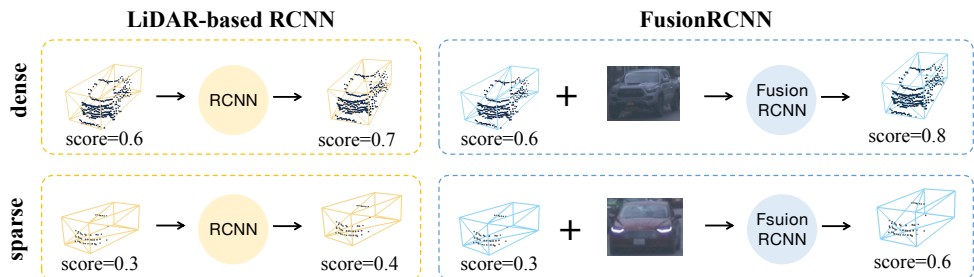

**Figure 1.** Comparison of our method with previous LiDAR-based two-stage methods. LiDAR-based methods often struggle to determine object categories and produce less confident scores correctly. The confidence score indicates the likelihood of an object (e.g., vehicle) being present in the box and the accuracy of the bounding box. Our approach, on the other hand, effectively integrates point cloud structure with dense image information, allowing us to overcome these challenges.

In this study, we focus on the refinement stage of fusing LiDAR point clouds and images. Previous approaches, such as those proposed by Xie et al. [11], have used an image segmentation sub-network to extract image features, which are then attached to the raw points. However, we have found that point-based fusion methods tend to ignore the semantic density of image features and heavily rely on the image segmentation sub-network. To address these limitations, we propose a new deep fusion method called FusionRCNN, which consists of three key steps: Firstly, RoI features are extracted from both points and images corresponding to proposals generated by any one-stage detectors. Since the number of point clouds and image sizes corresponding to different proposals can vary, we sample or pad points within the corresponding box to a unified number and employ RoIPooling to obtain RoI image features with a unified size. These methods can be used to extract RoI features in parallel, significantly improving model speed. Secondly, we fuse the features of the two modalities through well-designed intra-modality self-attention and inter-modality cross-attention mechanisms. Due to the differences in visual appearance and spatial structures between LiDAR and camera images, FusionRCNN first utilizes intra-modality self-attention to enhance domain-specific features before using cross-attention to fuse information from the two modalities dynamically. This approach abandons the heavy reliance on hard associations between points and images while maintaining the semantic density of images. Finally, we feed the encoded fusion features into a transformer-based decoder to predict refined 3D bounding boxes and confidence scores.

According to our observation, fully trained one-stage point cloud detectors have high bounding box recall rates (IoU(Intersection over Union) > 0.3) even in cases where the faraway point clouds are sparse. However, the real challenge is that the lack of structural information in sparse point clouds leads to low confidence, poor localization, and incorrect classification of these proposal boxes; for example, a car is misclassified as a bicycle with only several points within the proposal. Our novel two-stage fusion approach improves the precision of proposal boxes more accurately. Although some well-designed one-stage fusion methods [12,13] have achieved good performance, our method provides a new perspective for multi-modality fusion detection. We propose a two-stage plug-and-play

refinement approach that can be attached as an additional enhancement module after any conventional detector without redesigning a highly coupled and heavy network for each specific point cloud detector, bringing more flexibility.

Our FusionRCNN method is a versatile approach that can greatly enhance the accuracy of 3D object detection. Through extensive experiments on two widely used autonomous driving datasets, KITTI [14] and Waymo [15], we have shown that our method outperforms LiDAR-only methods, especially for challenging samples with sparse point clouds (such as samples in the Hard level on KITTI and samples in the 50 m−Inf range on Waymo). Notably, when our two-stage refinement network is applied to the baseline model SECOND [16], it improves the detection performance by a remarkable **11.88** mAP(mean Average Precision) in the range of ≥50 m (from 46.93 mAP to 58.81 mAP for vehicle detection) on the Waymo dataset.

To sum up, this work makes the following contributions:

- We propose a versatile and efficient two-stage multi-modality 3D detector, FusionRCNN. The detector combines image and point clouds within regions of interest and can enhance existing one-stage detectors with minor modifications.
- We introduce a novel transformer-based mechanism that enables the simultaneous achievement of attentive fusion between pixel and point sets, providing rich context and structural information.
- Our method demonstrates superior performance when compared to two-stage approaches on challenging samples that have sparse points in both the KITTI and Waymo Open Dataset.

## 2. Related Works

### 2.1. LiDAR-Based 3D Detection

LiDAR-based 3D detection methods can be broadly categorized into three groups: Voxel-based, Point-based, and Range View. Voxel-based detectors convert unstructured point clouds into regular 2D/3D grids [17,18], making it easy to apply conventional CNNs. The pioneering work MV3D [19] projects the point clouds onto 2D bird's-eye view grids and places numerous predefined 3D anchors for generating highly accurate 3D candidate boxes. This motivated the development of subsequent efficient bev (bird's-eye view) representation methods, such as VoxelNet [20], which applies mini PointNet [21] for voxel feature extraction, and SECOND [16], which introduces 3D sparse convolution to accelerate 3D voxel processing. For Point-based methods, PointNet and its variants [22] directly use the raw points as input and apply symmetric operators to address the unorderliness of point clouds. PointRCNN [1] and STD [2] segment foreground points with PointNet and generate proposals. 3DSSD [23] proposes a new sampling strategy for efficient computation. Range View detectors [24,25] represent LiDAR point clouds as dense range images, where pixels contain extra accurate depth information. Compared to other methods, Voxel-based detectors balance efficiency and performance, and we have chosen the Voxel-based detector as RPN(Region proposal network) in this paper.

### 2.2. Camera-Based 3D Detection

Compared to LiDAR sensors, cameras provide more comprehensive texture information at a lower cost and are widely used in applications such as autonomous driving. However, the absence of accurate depth information from cameras leads to a disparity between camera-based 3D detectors and LiDAR-based detectors, prompting researchers to strive for enhanced accuracy of camera-based 3D detectors and minimize the gap. FCOS3D [26] expands the 2D image detector [27] into the 3D domain by incorporating an additional 3D regression branch. Subsequent methods such as PGD [28] and EPro-PnP [29] have further improved the depth modeling capabilities. Unlike direct object detection in perspective view, DETR3D [30], PETR [31], and GraphDETR3D [32] employ DETR [33,34]-based methods to design detection heads and learnable object queries in 3D space. Inspired by LiDAR-based architecture, another stream of camera-only 3D perception models employ

a view transformer [35–38] to explicitly transfer camera features from a perspective view to a bird's-eye view. BEVDet [39], M2BEV [40] effectively build 3D object detectors based on LSS [35] and OFT [38]. CaDDN [41] and BEVDepth [42] incorporate additional supervision depth estimation in the view transformer to enhance performance. BEVDet4D [43], BEV-Former [44], and PETRv2 [45] leverage temporal information and significantly enhance single-frame methods.

### 2.3. LiDAR-Camera 3D Detection

In recent times, the integration of LiDAR and camera sensors in 3D detection has gained traction due to their complementary nature. LiDARs provide sparse point clouds with accurate depth information, whereas cameras provide high-resolution images with rich color and texture. Previous approaches such as MV3D [19] utilize LiDAR bev features to create 3D object proposals and project them onto multi-view images for RoI feature extraction. F-PointNet [46] lifts image proposals into a 3D frustum, yielding superior performance. Point-level fusion methods apply LiDAR-based detectors to raw foreground LiDAR points, and several successful methods, such as PointPainting [47], PointAugmenting [48], MVP [49], FusionPainting [50], and AutoAlign, have used input-level decoration. Other methods, such as DeepFusion [13] and Deep Continues Fusion [51], perform feature-level decoration. Recent works, such as TransFusion [12] and FUTR3D [52], have initialized object queries in 3D space and fused image features on the proposals. However, few works have focused on two-stage fusion networks. In this paper, we propose a novel framework that serves as a plug-and-play RCNN [53,54] module for existing detectors, thereby significantly boosting their performance.

### 2.4. Transformer for Object Detection

Since the introduction of DETR [33], which uses a transformer architecture to convert a small set of learned object queries into a set of predictions, the Transformer [55] has gained significant success in 2D object detection. The CNN backbone in DETR extracts image features, and follow-up works [34,56,57] introduce positional information into the object queries to speed up and stabilize training. Inspired by the success of the Transformer in 2D object detection, researchers have attempted to extend this mechanism to point cloud processing. For example, Votr [58] introduces a voxel-based Transformer backbone to capture large context information efficiently. SST [59] proposes a single-stride Transformer for 3D object detection that achieves impressive performance. SWFormer [60] and MSsVT [61] introduce window-shifting operations into point clouds to learn multi-scale features. Other works [12,13] apply the Transformer mechanism to fuse different modalities such as LiDAR and camera. CT3D [4] introduces a Channel-wise Transformer architecture to two-stage 3D object detection, which enables the capture of rich contextual information from point clouds. In this paper, we propose a LiDAR-Camera fusion paradigm in Regions of Interest using the Transformer architecture.

## 3. Method

Given $M$ predicted proposals containing 3D bounding boxes $B = \{b_i\}_{i=1}^{M}$, where $b_i = \{x, y, z, l, h, w, \theta\}$ denote the center position, size, and heading angle of the box, respectively, and confidence scores $S = \{s_i\}_{i=1}^{M}$ obtained from any one-stage detectors, we aim to enhance the detection results by leveraging point clouds $P$ and camera images $I = \{I_i \in \mathbb{R}^{3 \times H_I \times W_I}\}_{i=1}^{T}$ from $T$ different views, i.e.,

$$(B_r, S_r) = \mathcal{R}(B, P, I), \tag{1}$$

where $B_r$ and $S_r$ denote the corrected bounding boxes and confidence scores, respectively, and $\mathcal{R}$ represents the proposed network.

Figure 2 depicts the overall architecture of the proposed FusionRCNN. We employ the RoI Feature Extractor (Section 3.1) to extract the RoI features from the points and images corresponding to *B*, and then fuse the features of these two modalities via the Fusion Encoder (Section 3.2). The encoding fusion features are then fed into the Decoder (Section 3.3) to predict the refined 3D bounding boxes and confidence scores.

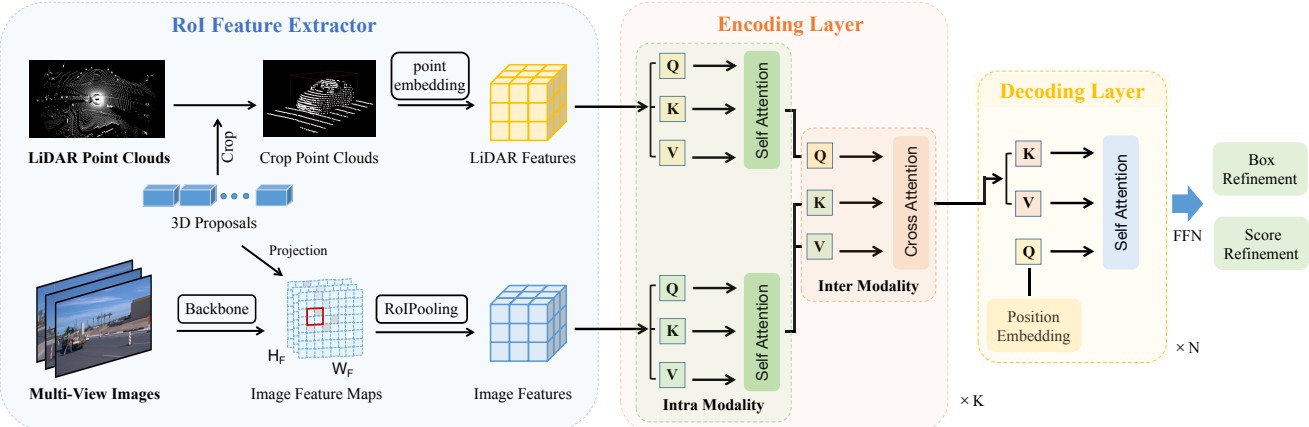

**Figure 2.** Overall architecture of FusionRCNN. Given 3D proposals, LiDAR and image features are extracted separately through the RoI feature extractor. Then, the features are fed into *K* fusion encoding layers which comprise self-attention and cross-attention modules. Finally, point features fused with image information are further fed into a decoder and predict the refined 3D bounding boxes and confidence scores.

### 3.1. RoI Feature Extractor

To capture enough structural and contextual information from 3D bounding boxes *B*, point clouds *P*, and camera images *I*, we keep the center of each bounding box $b_i$ fixed while expanding its length, width, and height by a ratio *k*. We then feed the scaled RoI to the feature extractor, using a two-branch architecture to extract the point/image RoI features individually from the point clouds *P* and images *I*.

The point branch involves sampling or padding points within the expanded box $b_i$ to a fixed number *N*. To enhance the point features, we follow the point embedding techniques utilized in [3,4] by concatenating the distances to the box's eight corners and its center, as well as additional LiDAR point information such as reflectivity:

$$F_i^P = \mathcal{L}(\Delta p_1, \Delta p_2, ..., \Delta p_8, p_b, p_e),\qquad(2)$$

where $\Delta p_j$ represents the distance to the *j*-th corner of the box $b_i$, $p_b$ denotes the center coordinates of the bounding box, $p_e$ contains extra LiDAR point information, and $\mathcal{L}(\cdot)$ is a linear projection layer that maps point features into an embedding with *C* channels. The resulting point RoI features are $F^P = \{F_i^P \in \mathbb{R}^{C \times N}\}_{i=1}^M$.

In the image branch, ResNet [62] and FPN [63] are used to convert the original multi-view images into feature maps. Next, we project the expanded 3D bounding boxes onto the 2D feature map and extract the image embedding corresponding to the RoI by cropping the 2D feature. Specifically, we project the eight 3D corners onto the 2D feature map using the intrinsic and extrinsic of the cameras. From this projection, we calculate the minimum circumscribed rectangle and perform RoIPooling to obtain the image feature $F_i^I$ with a unified size of $S \times S$ corresponding to $b_i$. Finally, another linear layer projects $F_i^I$ into the same dimension *C* as the point features. Formally, the image RoI features are $F^I = \{F_i^I \in \mathbb{R}^{C \times S \times S}\}_{i=1}^M$.

### 3.2. Fusion Encoder

Utilizing the RoI Feature Extractor described above, we can get the per-point feature and the per-pixel image feature (pixel size varies since we fix a $S \times S$ pooling size while the projected proposal sizes are different) inside the RoI. Instead of fusing features by painting the image features into points like previous methods [47,48], which prefer to utilize the direct correspondence between points and image pixels but neglects the fact that a local region of pixels can contribute to one point and vice versa, we leverage self-attention and cross-attention to achieve the Set-to-Set fusion.

### 3.2.1. Intra-Modality Self-Attention

To better model the inner relationships within each modality, we first feed point features and image features into the intra-modality self-attention layer. For embedded point features $F^P$, we have

$$Q_P, K_P, V_P = W_P^Q F^P, W_P^K F^P, W_P^V F^P, \tag{3}$$

$$F_{attn}^P = \text{LN}(\text{Attention}(Q_P, K_P, V_P) + F^P), \tag{4}$$

where $W_P^Q, W_P^K, W_P^V$ are linear projections and $\text{LN}(\cdot)$ represents layernorm layer. $\text{Attention}(\cdot)$ represents the multi-head attention, in which the results of $h$-th head can be obtained as

$$F_{attn} = \text{Softmax}(\frac{Q_h K_h^T}{\sqrt{d}}), \tag{5}$$

where $d$ is the feature dimension.

Correspondingly, the image features are fed into another multi-head self-attention layer to enhance the context information as

$$Q_I, K_I, V_I = W_I^Q F^I, W_I^K F^I, W_I^V F^I, \tag{6}$$

$$F_{attn}^I = \text{LN}(\text{Attention}(Q_I, K_I, V_I) + F^I). \tag{7}$$

### 3.2.2. Inter-Modality Cross-Attention

We combine the information from both domains by aligning point and image features at the feature level using inter-modality cross-attention. This is achieved as follows:

$$Q_{IP}, K_{IP}, V_{IP} = W_{IP}^Q F_{attn}^P, W_{IP}^K F_{attn}^I, W_{IP}^V F_{attn}^I, \tag{8}$$

$$F_{cross}^{PI} = \text{LN}(\text{Attention}(Q_{IP}, K_{IP}, V_{IP}) + F_{attn}^P), \tag{9}$$

It should be noted that the cross-attention mechanism is not mandatory, and the point and image branches can operate independently. This enhances the flexibility of our model and enables us to train the network in a decoupled manner.

Finally, $F_{cross}^{PI}$ are fed into FFN with two linear layers.

$$F^{PI} = \text{FFN}(F_{cross}^{PI}). \tag{10}$$

To enhance the complementary nature of the two modalities in the encoding layer, we employ a novel fusion strategy. This strategy involves integrating the rich semantic information of the image into the point features. Additionally, the object structure information extracted from the point branches is utilized to guide the aggregation of image features, which reduces the impact of occlusion and other situations. Our fusion encoder consists of multiple encoding layers to ensure complete feature fusion. The attention map visualization is presented in Figure 3.

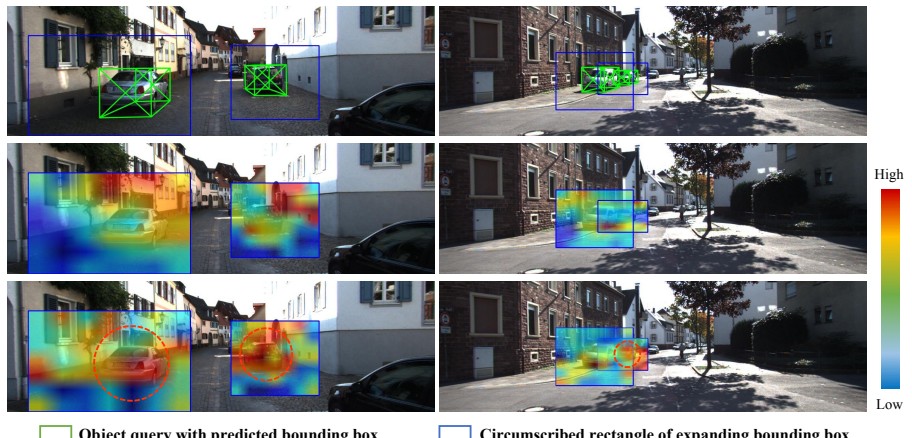

| Object query with predicted bounding box | Circumscribed rectangle of expanding bounding box |

**Figure 3.** Visualizations of attention map. First Row: input images and the predictions of object queries projected on the images (painted in green), and the circumscribed rectangle of expanding predictions projected on the images are painted in blue. Second Row: Intra-modality self-attention maps within the expanding RoI area of the image branch with high attention weights on the part of the vehicles and background. Third Row: Inter-modality cross-attention maps of image branch, higher attention weights are on the vehicles. Our fusion strategy can dynamically choose relevant image features as supplementary information with the help of the Intra-modality and Inter-modality attention modules. The two images are picked from the KITTI dataset.

### 3.3. Decoder

The encoded fusion features are then passed to the decoding layers to obtain the final box features. To achieve this, we start by initializing a learnable query embedding $E$ with $d$ channels as a query. The encoded features are then used as keys and values, as shown below:

$$Q_D, K_D, V_D = W_D^Q E, W_D^K F^{PI}, W_D^V F^{PI}, \tag{11}$$

Next, we perform layer normalization on the output of the attention mechanism added to the query embedding $E$, resulting in $E'$:

$$E' = \text{LN}(\text{Attention}(Q_D, K_D, V_D) + E), \tag{12}$$

Finally, we apply a feedforward neural network to $E'$, resulting in $E''$:

$$E'' = \text{FFN}(E'), \tag{13}$$

Here, $F^{PI}$ represents the output fusion features from the fusion encoding layers. The decoder module comprises multiple decoding layers.

### 3.4. Learning Objectives

To train our model, we adopt an end-to-end strategy. The overall loss function is the sum of the region proposal network (RPN) loss and the second stage network loss. The RPN loss is taken from the original network (SECOND [16]). The newly introduced second stage loss comprises the confidence loss $L_{conf}$ and the regression loss $L_{reg}$, as shown below:

$$L = L_{RPN} + L_{conf} + L_{reg}. \tag{14}$$

To guide the prediction of positive and negative samples, we employ the binary cross-entropy loss, which is defined as follows:

$$L_{conf} = -y \log(\hat{s}) - (1 - y) \log(1 - \hat{s}). \tag{15}$$

The division of positive and negative samples is based on the intersection over union (IoU) threshold as follows:

$$y = \begin{cases} 1, & \text{IoU} \geq t \\ 0, & \text{IoU} < t \end{cases}, \tag{16}$$

where $t$ is a threshold of IoU.

For positive samples, the regression loss $L_{reg}$ is composed of the smooth L1 loss of all parameters of the bounding box, as shown below:

$$L_{reg} = \sum_{p \in x,y,z,l,h,w,\theta} L_{smooth-L1}(\hat{p}, p), \tag{17}$$

where $\hat{p}, p$ represent the predicted and ground truth parameters of the bounding box, respectively.

## 4. Experiments

We assess the efficacy of FusionRCNN on two benchmark datasets: KITTI [14,64] and Waymo Open Dataset [15]. Moreover, we perform comprehensive ablation studies to substantiate the soundness of our design decisions.

### 4.1. Implementation Details

**Model setup.** Our network implementation is based on the open-sourced OpenPCDet [65] which is an open-source codebase based on PyTorch. We employ SECOND [16] as the Region Proposal Network (RPN), utilizing the settings established in OpenPCDet. In the RoI head, we utilize a ResNet50 image backbone pre-trained on ImageNet [66]. To save time, we freeze its weights during training and select the highest resolution output of FPN as the feature map. For each RoI, we set the expansion ratio $k$ to 2 and sample or pad 256-point clouds. The corresponding projected image region is converted to a $7 \times 7$ resolution by RoIPooling. Furthermore, we balance performance and efficiency by setting the number of encoding layers to 3 and the number of decoding layers to 1.

**Training details.** We train the network end-to-end on 8 Tesla V100 GPUs. For the Waymo Open Dataset, we employ the Adam optimizer and apply the cycle decay strategy with a learning rate of 0.0008. Consistent with CT3D [4], we train the model for 80 epochs. On KITTI, we adopt the same training strategy and train for 100 epochs with a learning rate of 0.003. Additionally, we utilize various forms of data augmentation, including flipping, rotation, and scaling, for both images and point clouds.

### 4.2. Results on Waymo

**Data and metrics.** Waymo Open Dataset is a large-scale outdoor public dataset for autonomous driving research, which contains RGB images from five high-resolution cameras and 3D point clouds from five LiDAR sensors. The whole dataset consists of 798 scenes (20 s fragment) for training and 202 scenes for validation, and 150 for testing. The measures are reported based on the distances from 3D objects to the sensor, i.e., 0–30 m, 30–50 m, and >50 m, respectively. These metrics are further divided into two difficulty levels: LEVEL_1 for 3D boxes with more than 5 LiDAR points and LEVEL_2 for boxes with at least 1 LiDAR point. Remarkably, the cameras in Waymo only cover around 250 degrees but not 360 degrees horizontally. Our framework can adapt to this situation, and the procedure of FusionRCNN is summarized in Algorithm A1. All models are trained on 20% Waymo dataset.

**Main results.** The performance of FusionRCNN on the Waymo Open Dataset is evaluated first. Table 1 presents the results of vehicle detection with 3D and BEV AP on the validation sequences. Notably, with the strong SECOND [16] baseline, FusionRCNN surpasses all previous methods in both LEVEL_1 and LEVEL_2, outperforming PV-RCNN [7] by 8.61% mAP and Voxel-RCNN [6] by 3.32% mAP on LEVEL_1. FusionRCNN achieves a 78.91% 3D mAP for the commonly used LEVEL_1 evaluation metric, which is a significant

improvement over the previous state-of-the-art method CT3D [4] by 2.61% mAP. We attribute this performance gain to our novel two-stage deep fusion design, which effectively integrates geometry information from LiDAR and dense texture information from the camera, thereby accurately refining bounding box parameters and confidence scores.

**Table 1.** Performance comparisons with state-of-the-art methods of vehicle detection on the Waymo dataset with 202 validation sequences (~40 k samples).*: re-implemented by ourselves on OpenPCDet. The top results are bolded in the table.

| Difficulty | Method | Reference | 3D Detection—Vehicle | | | | BEV Detection—Vehicle | | | |
|---|---|---|---|---|---|---|---|---|---|---|
| | | | Overall | 0–30 m | 30–50 m | 50 m-Inf | Overall | 0–30 m | 30–50 m | 50 m-Inf |
| LEVEL_1 | SECOND * [16] | *Sensor 2018* | 72.46 | 90.30 | 70.52 | 46.93 | 89.42 | 96.58 | 88.76 | 77.55 |
| | PointPillar [67] | *CVPR 2019* | 56.62 | 81.01 | 51.75 | 27.94 | 75.57 | 92.10 | 74.06 | 55.47 |
| | MVF [68] | *CoRL 2020* | 62.93 | 86.30 | 60.02 | 36.02 | 80.40 | 93.59 | 79.21 | 63.09 |
| | Pillar-OD [69] | *arXiv 2020* | 69.80 | 88.53 | 66.50 | 42.93 | 87.11 | 95.78 | 84.87 | 72.12 |
| | PV-RCNN [7] | *CVPR 2020* | 70.30 | 91.92 | 69.21 | 42.17 | 82.96 | 97.35 | 82.99 | 64.97 |
| | Voxel-RCNN [6] | *AAAI 2021* | 75.59 | 92.49 | 74.09 | 53.15 | 88.19 | 97.62 | 87.34 | 77.70 |
| | LiDAR-RCNN [3] | *CVPR 2021* | 76.00 | 92.10 | 74.60 | 54.50 | 90.10 | 97.0 | 89.50 | 78.90 |
| | Pyramid R-CNN [70] | *ICCV 2021* | 76.30 | **92.67** | 74.91 | 54.54 | - | - | - | - |
| | CT3D [4] | *ICCV 2021* | 76.30 | 92.51 | 75.07 | 55.36 | 90.50 | **97.64** | 88.06 | 78.89 |
| | **FusionRCNN (Ours)** | - | **78.91** | 92.38 | **77.82** | **58.81** | **91.94** | 97.12 | **91.22** | **85.22** |
| LEVEL_2 | SECOND * [16] | *Sensor 2018* | 64.14 | 89.04 | 64.14 | 35.98 | 82.23 | 95.63 | 83.26 | 64.29 |
| | PV-RCNN [7] | *CVPR 2020* | 65.36 | 91.58 | 65.13 | 36.46 | 77.45 | 94.64 | 80.39 | 55.39 |
| | Voxel-RCNN [6] | *AAAI 2021* | 66.59 | 91.74 | 67.89 | 40.80 | 81.07 | 96.99 | 81.37 | 63.26 |
| | LiDAR-RCNN [3] | *CVPR 2021* | 68.30 | 91.30 | 68.50 | 42.40 | 81.70 | 94.30 | 82.30 | 65.80 |
| | CT3D [4] | *ICCV 2021* | 69.04 | **91.76** | 68.93 | 42.60 | 81.74 | **97.05** | 82.22 | 64.34 |
| | **FusionRCNN (Ours)** | - | **70.33** | 91.22 | **71.47** | **46.21** | **84.39** | 96.22 | **86.15** | **70.18** |

Additionally, Table 2 presents the multi-class detection results of the map and mAPH (Mean average precision weighted by heading) for Vehicles, Pedestrians, and Cyclists on the Waymo Open Dataset. With the adoption of FusionRCNN, both the baseline model SECOND and CenterPoint [71] show significant improvements in detecting small objects, achieving 10.55% mAP on Cyclist for SECOND and 6.43% on Pedestrian for CenterPoint. Moreover, our method outperforms other single-frame methods in the stricter evaluation standard (IoU threshold of 0.8), as shown in Table 3, indicating that our approach performs well in accurately localizing objects with rich structure and texture information.

**Table 2.** Multi-class 3D detection results on Waymo validation set. Both SECOND and CenterPoint baselines are implemented in OpenPCDet. "+FusionRCNN" means that we add our FusionRCNN on the baseline detector. The performance improvements are painted in blue.

| Difficulty | Method | Vehicle | | Pedestrian | | Cyclist | |
|---|---|---|---|---|---|---|---|
| | | mAP | mAPH | mAP | mAPH | mAP | mAPH |
| LEVEL_1 | SECOND [16] | 70.96 | 70.34 | 65.23 | 54.22 | 57.13 | 55.62 |
| | +FusionRCNN | 77.67(+6.71%) | 77.10(+6.76%) | 70.63(+5.40%) | 61.88(+7.66%) | 67.55(+10.42%) | 66.17(+10.55%) |
| | CenterPoint [71] | 72.76 | 72.23 | 74.19 | 67.96 | 71.04 | 69.79 |
| | +FusionRCNN | 75.09(+2.33%) | 74.66(+2.43%) | 80.84(+6.65%) | 75.37(+7.41%) | 71.80(+0.76%) | 70.79(+1.00%) |
| LEVEL_2 | SECOND | 62.58 | 62.02 | 57.22 | 47.49 | 54.97 | 53.53 |
| | +FusionRCNN | 68.84(+6.26%) | 68.32(+6.30%) | 62.67(+5.45%) | 54.66(+7.17%) | 64.67(+9.70%) | 63.36(+9.83%) |
| | CenterPoint | 64.91 | 64.42 | 66.03 | 60.34 | 68.49 | 67.28 |
| | +FusionRCNN | 66.27(+1.36%) | 65.88(+1.46%) | 72.46(+6.43%) | 67.32(+6.98%) | 69.14(+0.65%) | 68.17(+0.89%) |

**Table 3.** Results on normal and strict IoU threshold. The normal and strict thresholds for vehicles are 0.7 and 0.8 on the Waymo validation set, respectively, ∗: results from [72]. The top results are bolded in the table.

| Method | Modality | Vehicle | |
|---|---|---|---|
| | | Normal | Strict |
| PointPillars [67] | L | 72.08 | 36.83 |
| PV-RCNN * [7] | L | 70.47 | 39.16 |
| MVF++ * [72] | L | 74.64 | 43.30 |
| SST [59] | L | 74.22 | 44.08 |
| **FusionRCNN(Ours)** | LC | **78.91** | **47.02** |

**Visualization.** The results on the Waymo dataset demonstrate that FusionRCNN performs better in long-range detection compared to CT3D, which merely uses point clouds in the refinement stage. The qualitative comparison in Figure 4 shows that CT3D has a comparable performance with FusionRCNN within a 50 m distance since point clouds nearby are dense enough. However, in the long-range distances, FusionRCNN outperforms CT3D. For instance, the three vehicles in red circles in the figure show that compared with the baseline SECOND, Proposal 1 and Proposal 2 have better locations and parameters with the refinement of CT3D, i.e., Proposal 1 is closer to ground truth due to a better yaw angle. CT3D and SECOND both fail to detect Proposal 3, yet FusionRCNN successes, we do an in-depth analysis about it. Proposal 3 only contains several point clouds resulting in the inability to determine the category accurately with a low confidence score, while FusionRCNN utilizes images branch with rich context information, which makes it easy to judge the category from a camera view.

*4.3. Results on KITTI*

**Data and metrics.** KITTI Dataset has been widely used in 3D detection tasks since its release. It contains multiple types of sensors like stereo cameras and a 64-beam Velodyne. There are 7481 training samples commonly divided into 3712 samples for training and 3769 samples for validation, and 7518 samples for testing. We conduct experiments on the commonly used category car whose detection IoU threshold is 0.7. We also report the results for three difficulty levels (easy, moderate, and hard) according to the object size, occlusion state, and truncation level.

**Main results.** To validate our framework, we conducted experiments on the KITTI validation set and compared its performance with previous state-of-the-art methods. As shown in Table 4, our FusionRCNN significantly outperforms the one-stage method SECOND for all three difficulty levels, with improvements of +1.29% for Easy, +7.02% for Moderate, and +2.1% for Hard. Additionally, it shows great competitiveness with all LiDAR-based and LiDAR-Camera methods. Moreover, our FusionRCNN performs better than the two-stage fusion competitor PI-RCNN [11], with an improvement of 7.11% on Moderate mAP. We also compared FusionRCNN with the released methods PV-RCNN [7] and CT3D [4], as they share the same RPN. Our FusionRCNN performs better than PV-RCNN in all difficulty levels, and when compared with the state-of-the-art method CT3D, our method shows better overall performance, leading CT3D by 0.36% on the Easy level, 0.39% on the Moderate level, and 0.33% on the Hard level. Remarkably, our FusionRCNN achieves an AP of 79.32% (Hard) and outperforms state-of-the-art 3D detectors. For further validation, we report comparisons with previous methods on the KITTI test set in Table 5. FusionRCNN achieves better performance than other competitive methods on the Moderate and Hard levels, leading the state-of-the-art method CT3D by 0.36%, 0.39%, and 0.33% on Easy, Moderate, and Hard levels, respectively. The experiments indicate that Our novel two-stage fusion framework better captures structural and contextual information effectively compared with point-based two-stage methods.

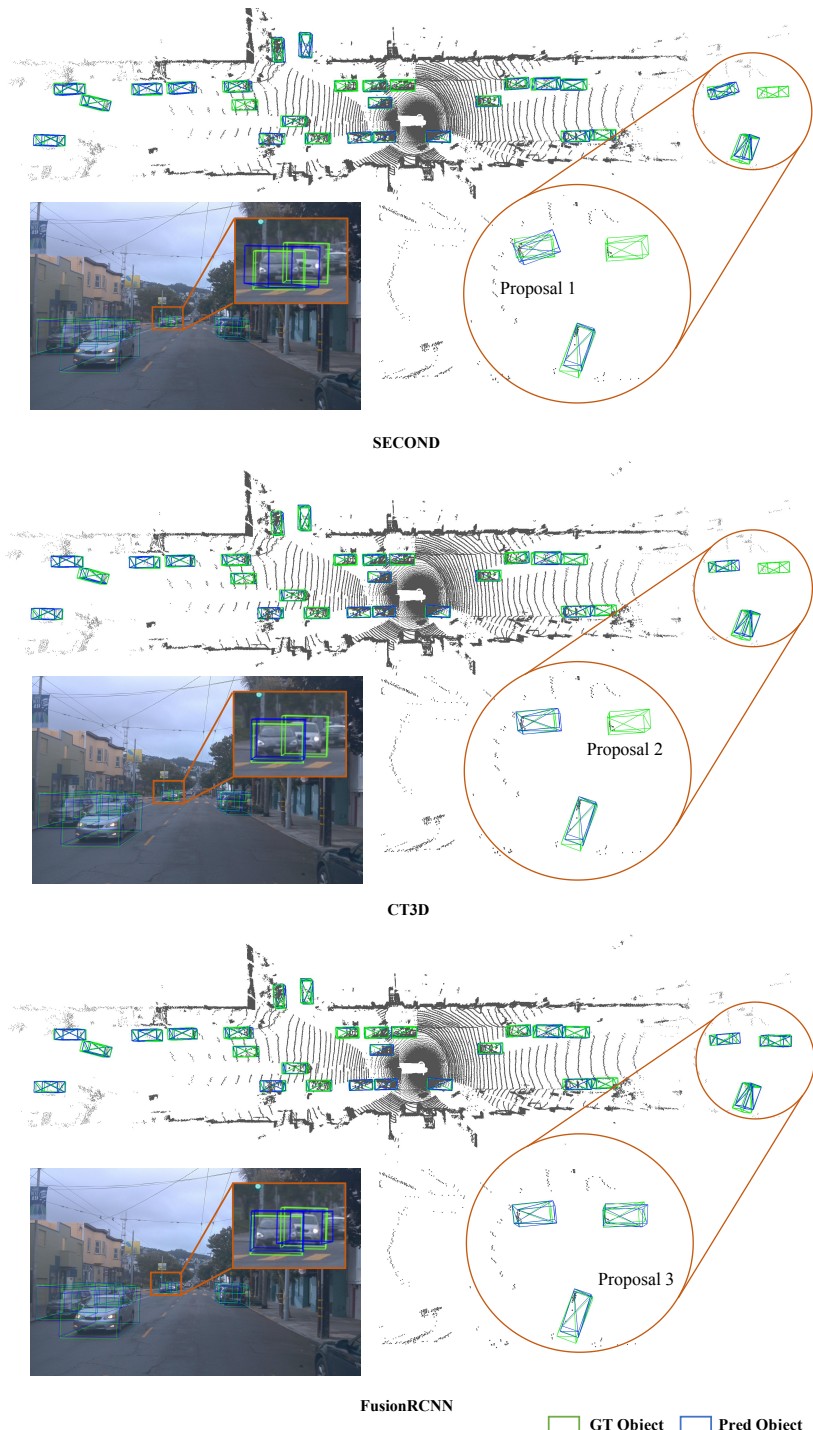

**Figure 4.** Qualitative comparison between LiDAR-based two-stage detector (CT3D) and our Fusion-RCNN on the Waymo Open Dataset. Ground-truth and predictions are painted in Green and Blue, respectively. Three proposal vehicles in red circles are zoom-in and visualized on 2D images and 3D point clouds. Our FusionRCNN works better than CT3D with only LiDAR input in long-range detection.

**Table 4.** Results on KITTI *val*. AP(Average Precision) with 0.7 IoU threshold and 11 recall positions are reported. L denotes LiDAR-only approaches. LC denotes the methods that combine LiDAR point clouds and camera images. The best results are bolded and performance improvements are painted in blue.

| Method | Modality | 3D Detection—Car | | |
|---|---|---|---|---|
| | | Easy | Mod. | Hard |
| MV3D [19] | LC | 71.29 | 62.68 | 56.56 |
| ContFuse [51] | LC | - | 73.25 | - |
| AVOD-FPN [73] | LC | - | 74.44 | - |
| F-PointNet [46] | LC | 83.76 | 70.92 | 63.65 |
| PI-RCNN [11] | LC | 88.27 | 78.53 | 77.75 |
| 3D-CVF at SPA [74] | LC | 89.67 | 79.88 | 78.47 |
| PointPillars [67] | L | 86.62 | 76.06 | 68.91 |
| STD [2] | L | 89.70 | 79.80 | 79.30 |
| PointRCNN [1] | L | 88.88 | 78.63 | 77.38 |
| SA-SSD [75] | L | **90.15** | 79.91 | 78.78 |
| 3DSSD [23] | L | 89.71 | 79.45 | 78.67 |
| PV-RCNN [7] | L | 89.35 | 83.69 | 78.70 |
| Voxel-RCNN [6] | L | 89.41 | 84.52 | 78.93 |
| Pyramid R-CNN [70] | L | 89.37 | 84.38 | 78.84 |
| CT3D [4] | L | 89.54 | 86.06 | 78.99 |
| SECOND [16] | L | 88.61 | 78.62 | 77.22 |
| **+FusionRCNN (Ours)** | LC | 89.90 (**+1.29%**) | **86.45 (+7.93%)** | **79.32 (+2.10%)** |

**Table 5.** Performance comparison on the KITTI test set. The results are reported with AP calculated by 40 recall positions and 0.7 IoU threshold for the car class. L denotes LiDAR-only approaches. LC denotes the methods that combine LiDAR point clouds and camera images. The best results are bolded and performance improvements are painted in blue.

| Method | Modality | 3D Detection—Car | | |
|---|---|---|---|---|
| | | Easy | Mod. | Hard |
| MV3D [19] | LC | 74.97 | 63.63 | 54.00 |
| ContFuse [51] | LC | 83.68 | 68.78 | 61.67 |
| AVOD-FPN [73] | LC | 83.07 | 71.76 | 65.73 |
| F-PointNet [46] | LC | 82.19 | 69.79 | 60.59 |
| PI-RCNN [11] | LC | 84.37 | 74.82 | 70.03 |
| 3D-CVF at SPA [74] | LC | 89.20 | 80.67 | 77.15 |
| PointPillars [67] | L | 82.58 | 74.31 | 68.99 |
| STD [2] | L | 87.95 | 79.71 | 75.09 |
| PointRCNN [1] | L | 86.96 | 75.64 | 70.70 |
| SA-SSD [75] | L | 88.75 | 79.79 | 74.16 |
| 3DSSD [23] | L | 88.36 | 79.57 | 74.55 |
| PV-RCNN [7] | L | 90.25 | 81.43 | 76.82 |
| Voxel-RCNN [6] | L | **90.90** | 81.62 | 77.06 |
| CT3D [4] | L | 87.83 | 81.77 | 77.16 |
| SECOND [16] | L | 83.34 | 72.55 | 65.82 |
| **+FusionRCNN (Ours)** | LC | 88.12 (**+4.78%**) | **81.98 (+9.43%)** | **77.53 (+11.71%)** |

**Visualization.** Experiments on KITTI show that our method performs excellently in Moderate and Hard level detection. We also show a qualitative comparison between FusionRCNN and CT3D and the comparison is shown in Figure 5.

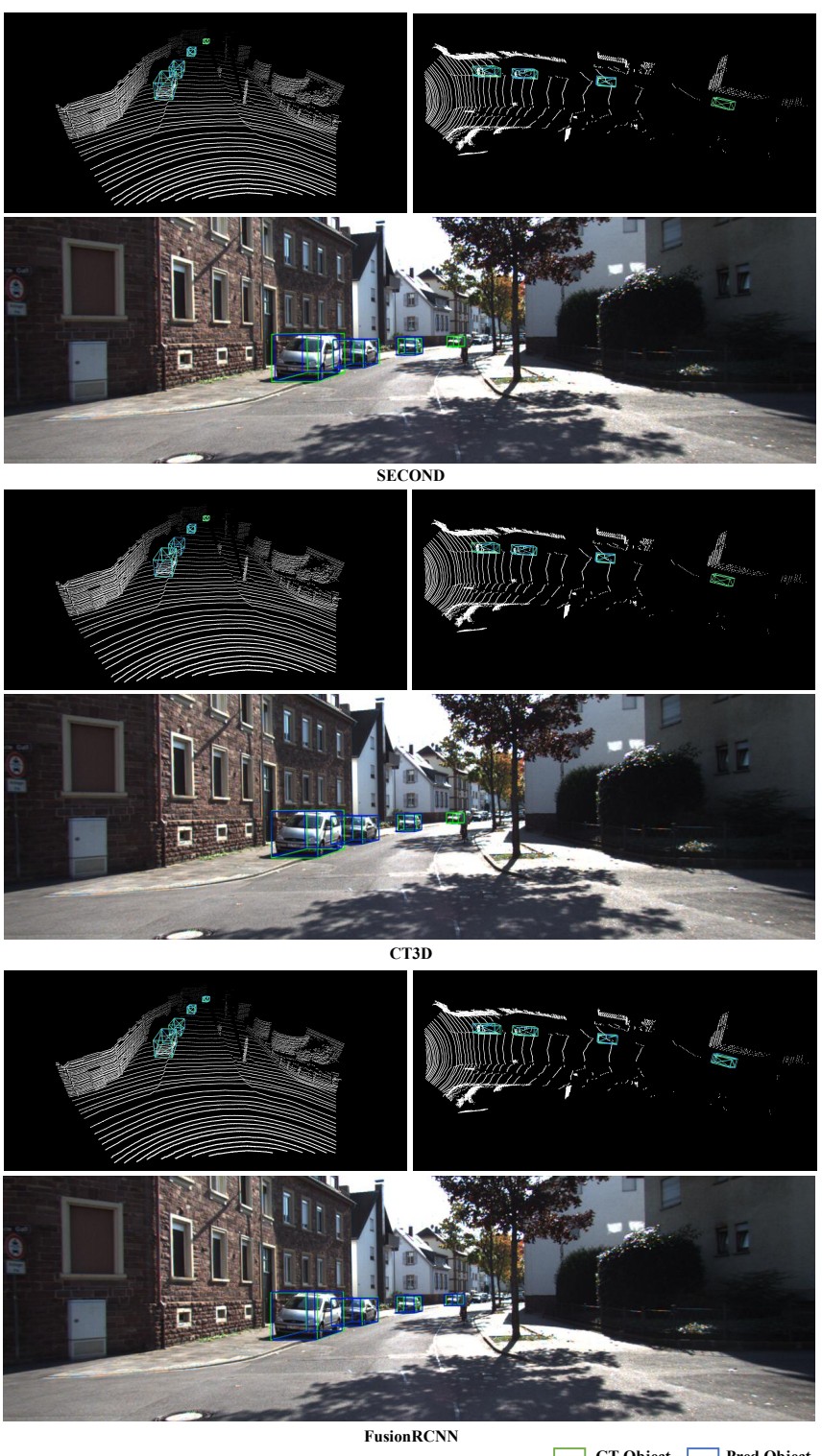

**Figure 5.** Qualitative comparison between LiDAR-based two-stage detector (CT3D) and our FusionR-CNN on the KITTI Dataset. Green boxes and Blue boxes are ground-truth and predictions, respectively. Our FusionRCNN performs better than CT3D with only LiDAR input in long-range detection.

### 4.4. Ablation Studies

**Effect of LiDAR-Camera fusion.** We examine the impact of incorporating texture information from camera images on our detection method. To this end, we transform our FusionRCNN approach into a LiDAR-based technique, termed FusionRCNN-L, by

introducing an intra-modality self-attention mechanism for the image branch and an inter-modality cross-attention module in the Fusion Encoder. A comparison of the frameworks of FusionRCNN and FusionRCNN-L is illustrated in Figure 6, and we perform inference using identical settings for both methods. As depicted in Table 6, FusionRCNN-L achieves an mAP of 90.25% in Vehicle BEV detection and outperforms most of the techniques listed in Table 1. By leveraging LiDAR-Camera fusion, FusionRCNN exhibits further improvements, particularly for long-range detection (50 m-Inf). Our FusionRCNN approach aims to compensate for the limitations of sparse point clouds by introducing complementary image information. Our fusion strategy, depicted in Figure 3, can dynamically select relevant image features as supplementary information using the Intra-modality and Inter-modality attention module. Moreover, the structure information of the point cloud can guide the aggregation of image features to focus on the foregrounds, even in obscured situations (Third Row Right in Figure 3).

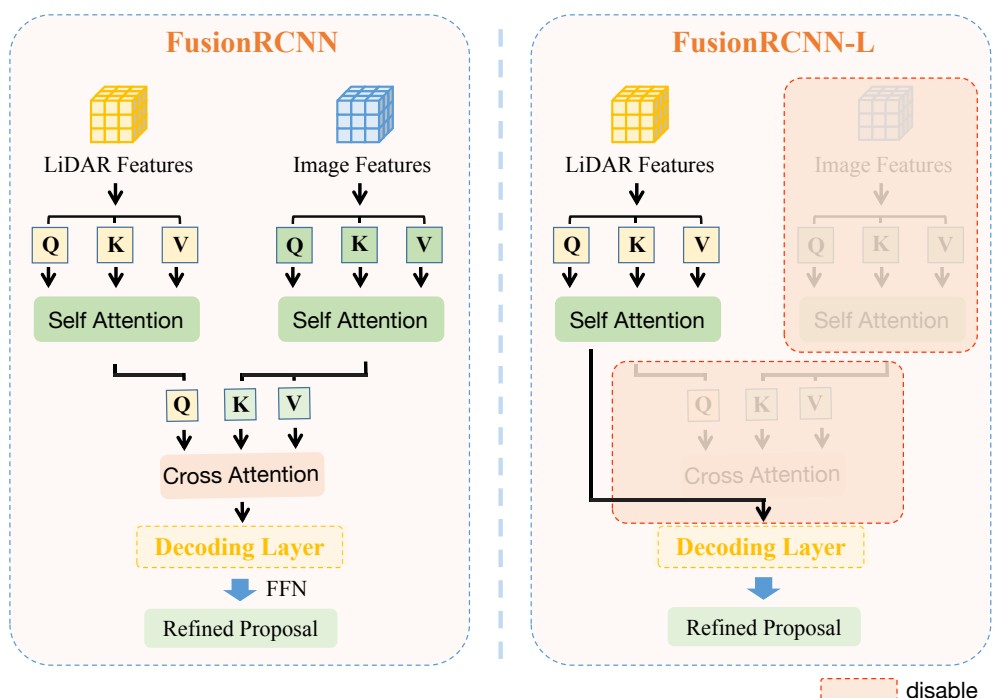

**Figure 6.** Framework comparison of FusionRCNN and FusionRCNN-L. FusionRCNN-L is a LiDAR-based method that disables intra-modality self-attention of the image branch and inter-modality cross-attention module in Fusion Encoder.

**Table 6.** Vehicle BEV detection under different distance on Waymo validation set. FusionRCNN-L donates the LiDAR-based two-stage method by disabling the image branch. The performance improvements and latency increase are painted in blue and red respectively.

| Method | Overall | 0–30 m | 30–50 m | 50 m-Inf | Latency (ms) |
|---|---|---|---|---|---|
| FusionRCNN-L | 90.25 | 96.58 | 89.24 | 80.61 | 125 |
| FusionRCNN | 91.94 (**+1.69%**) | 97.12 (**+0.54%**) | 91.22 (**+1.98%**) | 85.22 (**+4.61%**) | 185 (**+60**) |

**Different RPN Backbones.** To investigate the applicability of FusionRCNN to other single-stage detectors, namely SECOND, PointPillar, and CenterPoint, we integrated FusionRCNN into these baseline models. As shown in Table 7, our approach outperforms all three baseline models with significant gains, achieving a +6.14%, +2.7%, and +5.55% improvement in 3D mAP on LEVEL_1. This improvement stems from the effective utilization of our novel LiDAR-Camera fusion mechanism, which leverages the complementary information from the structure and semantic features extracted from both LiDAR and camera images.

**Table 7.** Ablations on different one-stage detectors on Waymo validation set. +FusionRCNN donates plugging FusionRCNN into different popular single-stage detectors. The performance improvements are painted in blue.

| Methods | LEVEL_1 | | LEVEL_2 | |
| --- | --- | --- | --- | --- |
| | 3D AP | APH | 3D AP | APH |
| SECOND [16] | 72.46 | 71.87 | 64.14 | 63.60 |
| +FusionRCNN | 78.91 (+6.45%) | 78.39 (+6.52%) | 70.65 (+6.51%) | 70.16 (+6.56%) |
| PointPillar [67] | 72.27 | 71.69 | 63.85 | 63.33 |
| +FusionRCNN | 74.67 (+2.40%) | 74.10 (+2.41%) | 65.96 (+2.11%) | 65.44 (+2.11%) |
| CenterPoint [71] | 72.08 | 71.53 | 63.55 | 63.06 |
| +FusionRCNN | 77.63 (+5.55%) | 77.16 (+5.63%) | 69.26 (+5.71%) | 68.83 (+5.77%) |

**RoI Feature Extractor.** Our RoI feature extractor contains a point and an image branch. Previous studies [3,4,7] have demonstrated that raw points contain more precise structure information, which is beneficial for extracting local bounding box contextual information. In this study, we mainly focus on investigating the effect of the image branch on our approach. We examine the impact of different parameters that may affect image feature extraction and, consequently, detection performance. Specifically, we experiment with various output sizes ($S$) of RoI image features and report the results in Table 8. We find that these settings have little impact on the image extraction branch. One possible explanation is that in our fusion encoding layer, the LiDAR and image features fuse dynamically, and the image features contribute to category classification with high-level contextual information. To extract each RoI image feature with the same size for parallel computing, we employ RoIPooling/RoIAlign. We conduct an ablation study to investigate the influence of these two operations and expansion radio on our approach, and the results are presented in Table 9. FusionRCNN achieves better performance When expansion radio is set large (e.g., k = 2.0), and there is little difference in performance between the two operations. This is attributed to the expansion of RoI that can effectively retain the foreground images corresponding to the proposals, ensuring that use texture information can be utilized via LiDAR-camera fusion. When the expansion ratio is small, RoIAglin performs better due to its more accurate RoI feature extraction ability, especially for far objects.

**Table 8.** Ablation on output size of RoI image features. the results are tested in the Waymo dataset with 202 validation sequences. The best results are bolded in the table.

| Output Size | LEVEL_1 3D AP/APH | LEVEL_2 3D AP/APH |
| --- | --- | --- |
| 3 × 3 | 78.88/78.36 | 70.63/70.14 |
| 5 × 5 | 78.82/78.30 | 70.57/70.10 |
| 7 × 7 | **78.91/78.39** | **70.65/70.16** |
| 9 × 9 | 78.87/78.37 | 70.62/70.13 |

**Table 9.** Ablations on RoI operations and expansion radio for image features. We report the results on the Waymo dataset with 202 validation sequences. The best results are bolded in the table.

| Expansion Ratio | Operation | LEVEL_1 3D AP/APH | LEVEL_2 3D AP/APH |
| --- | --- | --- | --- |
| k = 1.2 | RoIAlign | 78.47/77.95 | 69.88/69.71 |
| | RoIPooling | 78.41/77.88 | 69.81/69.63 |
| k = 1.5 | RoIAlign | 78.62/78.09 | 70.11/69.85 |
| | RoIPooling | 78.63/78.11 | 70.36/69.87 |
| k = 2.0 | RoIAlign | 78.83/78.31 | 70.57/70.11 |
| | **RoIPooling** | **78.91/78.39** | **70.65/70.16** |

## 5. Conclusions and Future Work

This work introduces a two-stage multi-modality 3D detector called FusionRCNN that combines LiDAR point cloud and camera image information in regions of interest. The proposed detector employs a sophisticated attention mechanism that enables Set-to-Set fusion, which enhances its robustness against LiDAR-Camera calibration noise. Our experiments demonstrate that FusionRCNN outperforms state-of-the-art two-stage 3D detectors on both Waymo Open Dataset and KITTI dataset. The proposed method is plug-and-play and has the potential to improve existing one-stage 3D detectors significantly. Our method provides accurate detection and can be used in tasks that require high precision, such as 3D offline auto-labeling. However, the severe occlusion and truncation at close range may lead to limited performance to some extent. In future work, we will explore how distance can be used as a parameter to optimize the model or add logical post-processing to improve performance further.

**Author Contributions:** Methodology and writing, X.X.; validation, S.D.; writing—original draft preparation, L.D.; visualization, J.W.; resources, and funding acquisition, T.X.; data curation, L.S.; data curation, P.J.; writing—review and editing, J.L. All authors have read and agreed to the published version of the manuscript.

**Funding:** This work was financially supported by the National Natural Science Foundation of China (No. 62101032), the Postdoctoral Science Foundation of China (Nos. 2021M690015, 2022T150050), and the Beijing Institute of Technology Research Fund Program for Young Scholars (No. 3040011182111).

**Data Availability Statement:** Two Publicly available datasets were analyzed in this paper. Waymo Open Dataset can be found here: https://waymo.com/open/ (accessed on 12 January 2023). KITTI dataset can be found at https://www.cvlibs.net/datasets/kitti/ (accessed on 12 January 2023).

**Conflicts of Interest:** The authors declare no conflict of interest.

## Abbreviations

The following abbreviations are used in this manuscript:

| | |
|---|---|
| 3D | Three dimensional |
| LiDAR | Light detection and ranging |
| BEV | Bird's eye view |
| RoI | Regions of Interest |
| mAP | Mean average precision |
| AP | Average precision |
| mAPH | Mean average precision weighted by heading |
| IoU | Intersection over union |
| RPN | Region proposal network |
| GT | Ground truth |

## Appendix A

---

**Algorithm A1** Proposed RoI Feature Extractor in FusionRCNN

---

**Input:** Giving 3D bounding boxes $B$; Point clouds $P$; Images feature maps $I$;
**Output:** Point RoI features $F^P$, Image RoI features $F^I$

  1: expand $B$ by radio k;
  2: **for** each box $b_i$ in $B$ **do**                                          ▷ LiDAR branch
  3:      sampled or padded points $P$ within $b_i$ to a unified number $N$
  4:      point embedding via Equation (2)
  5:      linear projection and get Point RoI Feature $F_i^P$
  6:      project $b_i$ from LiDAR coordinate to Image coordinates             ▷ Camera branch
  7:      **if** $b_i$ shows in Image coordinates **then**
  8:          perform RoI pooling to get the image feature $I$ with a unified size S × S
  9:          linear projection and get Image RoI Feature $F_i^I$
10:      **else**
11:          get the pseudo-zero-pad RoI feature with a unified size S × S
12:          linear projection and get image RoI feature $F_i^I$
13:      **end if**
14: **end for**
15: **return** return Point RoI features $F^P$, Image RoI features $F^I$

---

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
