# Peer review of "FusionRCNN: LiDAR-Camera Fusion for Two-Stage 3D Object Detection"

_remotesensing, doi:10.3390/rs15071839_

Round 1
Reviewer 1 Report
Notes:
Figure 1 – should be useful to explain what does score mean.
Rigo 63: it might be useful to spend a few words on a brief description of the two types of datasets.
Tigo 66: Please use the full name when entering an acronym for the first time
Table 1: what are the reasons why the performance at 0-30 meters is lower? Do you have any ideas to improve it in the future, even if the overall performance is still very high?
Why did you think about doing this research work? For what purpose ?
Check if in any formula it is possible to put a unit of measurement.
Maybe some sentences are not enough clear in English, and the conclusion support the research.
The figure and the table are are clear and understandable and are reported throughout the text.
The software used is not mentioned anywhere. Could it be useful to mention it?
The authors have chosen recent papers. They never went before 2017.
I have not found connection with the keyword "thematic mapping"
Author Response
\documentclass[11pt]{article}
\usepackage{color, latexsym, times, verbatim, amsmath, graphics,epsfig,array, amssymb, colordvi}
\usepackage{algorithm}
\usepackage{color, xcolor}
%\renewcommand{\baselinestretch}{2.0} %1.3
%\setlength{\oddsidemargin}{-0.7cm}
\setlength{\textwidth}{13.4625cm}
%\setlength{\evensidemargin}{-0.7cm}
\setlength{\columnsep}{0.9525cm}
\setlength{\textheight}{22.5425cm} \setlength{\hoffset}{0cm}
\setlength{\voffset}{0cm}\setlength{\topmargin}{-0.5cm}
%%%%%%%%%%%%%%%%%%%%%%%%%%%%%%%%%%%%%%%%%%%%%%%%%%%%%%%%%%%%%%%
\addtolength{\oddsidemargin}{-1.5cm}
\addtolength{\evensidemargin}{-2cm} \textwidth=16cm
\renewcommand{\baselinestretch}{1.2}
\begin{document}
\begin{center}
{\bf \LARGE {Responses to Reviewer 1}}
\end{center}
\large
\noindent\textbf{Manuscript ID:} remotesensing-2266845
\noindent\textbf{Title:} FusionRCNN: LiDAR-Camera Fusion for Two-stage 3D Object Detection
\noindent\textbf{Author:} Xinli Xu, Shaocong Dong, Tingfa Xu, Lihe Ding, Jie Wang, Peng Jiang, Liqiang Song, Jianan Li *
\\
\noindent We would like to thank you very much for your valuable comments and good suggestions that greatly helped to improve our manuscript. We have carefully considered your valuable comments and good suggestions. In the following, we are going to explain how your comments have been taken into full account in the revision. All modifications in the manuscript have been highlighted in \textcolor{yellow}{yellow}.
\\
% \noindent\textbf{1.} \underline{\textbf{Point:}} Figure 1 – should be useful to explain what does score mean.
% \\
\noindent{\textbf{[Point1]}} Figure 1 – should be useful to explain what does score mean.
\noindent{\textbf{Response:}}
Thank you for your kind reminders. We revised the sentence as follows:\\
{"The confidence score indicates the likelihood of an object (e.g., vehicle) being present in the box and the accuracy of the bounding box."[Caption of Figure1]}\\
\noindent{\textbf{[Point2]}} Rigo 63: it might be useful to spend a few words on a brief description of the two types of datasets.
\noindent{\textbf{Response:}}
Thank you for your kind suggestion. We revised the sentence as follows:\\
{"Through extensive experiments on \textbf{two widely used autonomous driving datasets}, KITTI and Waymo, we have shown that our method outperforms LiDAR-only methods, especially for challenging samples with sparse point clouds (such as samples in the Hard level on KITTI and samples in the $50m\mathrm{-Inf}$ range on Waymo)."[Line77]}\\
\noindent{\textbf{[Point3]}} Tigo 66: Please use the full name when entering an acronym for the first time.
\noindent{\textbf{Response:}}
Thank you for your kind reminders. We used the full name for the first time and revised the sentence as follows:\\
"Notably, when our two-stage refinement network is applied to the baseline model SECOND, it improves the detection performance by a remarkable 11.88 \textbf{mAP (mean Average Precision)} in the range of $ \ge 50m $ (from 46.93 mAP to 58.81 mAP for vehicle detection) on the Waymo dataset"[Line81]\\
"Compared to other methods, Voxel-based detectors balance efficiency and performance, and we have chosen the Voxel-based detector as \textbf{RPN (Region proposal network)} in this paper."[Line112]\\
"This motivated the development of subsequent efficient \textbf{bev (bird's-eye view)} representation methods, such as VoxelNet, which applies mini PointNet for voxel feature extraction, and SECOND, which introduces 3D sparse convolution to accelerate 3D voxel processing."[Line102]\\
"According to our observation, fully trained one-stage point cloud detectors have high bounding box recall rates (\textbf{IoU (Intersection over Union)}\textgreater0) even in cases where the distant point clouds are sparse."[Line65]\\
"\textbf{AP(Average Precision)} with 0.7 IoU threshold and 11 recall positions are reported."[Table4]\\
"Additionally, Table 2 presents the multi-class detection results of map and \textbf{mAPH (Mean average precision weightd by heading)} for Vehicle, Pedestrian, and Cyclist on the Waymo Open Dataset."[Line301]\\
\noindent{\textbf{[Point4:]}} Table 1: what are the reasons why the performance at 0-30 meters is lower? Do you have any ideas to improve it in the future, even if the overall performance is still very high?\\
\noindent{\textbf{Response:}}
Thank you for your careful observation and pointing out this problem. After in-depth analysis and visualization, we believe that it is mainly caused by the following reasons:\\
1. Truncation. Objects within 30 meters have a higher probability of appearing at the edge of the camera’s field of view, resulting in capturing part of the objects in the image. The incomplete image information may affect location accuracy.\\
2. Obscuration. Objects nearby are more likely to be occluded, and the image informations within the corresponding proposals may cause incorrect classification.\\
Although intra-modality self-attention and inter-modality cross-attention mechanisms alleviate the occlusion and truncation problems to some extent, as shown in Fig6. However, incorrect or incomplete texture information from images caused by severe occlusion and truncation affects the location and classification, e.g., a large truck blocks a vehicle. In follow-up work we believe that above problems can be further solved by a range-aware fusion mechanism. We can mainly rely on structure information from point clouds in close range as point clouds are dense enough, and utilize texture and structure information from sparse point clouds and images to jointly classify and locate foreground objects. Moreover, the application of logical post-processing can also be useful. [We summary above in conclusion and future work,Line402-407]\\
\noindent{\textbf{[Point5:]}} Why did you think about doing this research work? For what purpose?\\
\noindent{\textbf{Response:}}
Thank you for your comments.\\
% \textcolor{cyan}
{This work is derived from observations in practical applications. We find the sparsity of point clouds, especially for the points far away, making it difficult for the LiDAR-only refinement module to accurately recognize and locate objects. However, it is easy to judge from the images captured by cameras, as show in Figure1. To address this problem, we propose a novel multi-modality two-stage approach named \textbf{FusionRCNN}, which effectively and efficiently fuses point clouds and camera images in the Regions of Interest (RoI). Our method provides more accurate and robust detection, which can be used in autonomous driving and robotics. Since our method is plug-and-play and effectively improves the performance of the baseline model, especially for far objects. This work has great potential to be used in detection tasks that require high accuracy, such as 3D offline automatic labeling and 3D online detection. [We summary above in conclusion and future work,Line402-407]}\\
\noindent{\textbf{[Point6]}} Check if in any formula it is possible to put a unit of measurement.
\noindent{\textbf{Response:}}
Thank you for your kind suggestion. We have checked and done if appropriate.\\
\noindent{\textbf{[Point7]}} Maybe some sentences are not enough clear in English, and the conclusion support the research.
\noindent{\textbf{Response:}}
Thank you for your kind reminders. In the revised version, we have improved the expression of many sentences in the manuscript to make it easier to understand.{[All changes are highlighted in yellow in the revision]}\\
\noindent{\textbf{[Point8]}} The figure and the table are are clear and understandable and are reported throughout the text.
\noindent{\textbf{Response:}}
Thank you very much, we will continue to improve our work if you have any other questions.\\
\noindent{\textbf{[Point9]}} The software used is not mentioned anywhere. Could it be useful to mention it?
\noindent{\textbf{Response:}} Thank you for your kind reminders. We revised the sentence as follows:\\
"Our network implementation is based on the open-sourced OpenPCDet which is an open-source codebase based on PyTorch."[Line265]\\
\noindent{\textbf{[Point10]}} The authors have chosen recent papers. They never went before 2017.
\noindent{\textbf{Response:}}
Thank you for your kind reminders. As typical methods have been proposed in recent years, little works before 2017 were mentioned. In the new version I have added some methods before 2017 in Related Works.[Line99]\\
\noindent{\textbf{[Point11]}} I have not found connection with the keyword "thematic mapping"
\noindent{\textbf{Response:}}
Thank you for your kind reminders. This paper concerns on the problem of LiDAR-camera fusion for 3D detection in urban scenarios, e.g., autonomous driving and robotics. We propose a novel \textbf{multi-modality} two-stage approach which effectively and efficiently fuses point clouds and camera images in the Regions of Interest (RoI). While the special issue “Data Fusion for Urban Applications” of the journal “Remote Sensing” is also concerning on the \textbf{2-D, 3-D and multi-dimensional data fusion} for urban analysis, thus the paper is fitting the scope of the journal in the our consideration.\\
Finally, thank you so much again for all your valuable comments and suggestions. We hope the manuscript after careful revisions has addressed all the concerns and meet your high standards. We welcome further constructive comments if any.
\end{document}

Reviewer 2 Report
The manuscript introduces a LiDAR-camera fusion method of the 3D object detection task. This is a popular topic in computer vision research, with many works, such as FUTR3D[1] and TransFusion[2], focusing on it. In this paper, authors propose a two-stage multi-modal fusion mechanism and provide extensive experiment on popular autonomous driving datasets, such as KITTI and Waymo. Compared to the related work TransFusion[2], this paper performs multi-modal fusion in the second stage and observes significant improvement against the baseline. However, there are minor differences in performing fusion in one-stage manner or two-stage manner, thus the novelty of this paper is limited. I hope the contributions in the next edition will be enriched. Here are some issues in detail:
1. In general, this work extends prior multi-modal works to a two-stage manner. It uses 3D proposals provided by LiDAR detectors and fuses camera features in the second stage with the popular attention mechanism. However, proposals generated by the first stage don’t use the multi-modal information, if the LiDAR points are too far or sparse, the proposal will not be detected as positive, so it’s perplexing that why not perform multi-modal fusion in the first stage. I think the consideration of this design choice should be clarified in this paper.
2. All experiments in this article were conducted on the validation datasets, making it hard to compare with models in the official leaderboards. A better choice is to attach the test set results in the next edition.
3. In Table 8, it compares the model with the RoIAlign operation and with the RoIPooling operation, these results show a counterintuitive conclusion, however, this paper doesn’t give a detailed explanation. In general, RoIAlign performs better than RoIPooling, but table 8 shows quite the opposite. That’s an interesting point, please conduct more experiments and give some analysis.
Reference
[1] Chen, Xuanyao, et al. "Futr3d: A unified sensor fusion framework for 3d detection." arXiv preprint arXiv:2203.10642(2022).
[2] Bai, Xuyang, et al. "Transfusion: Robust lidar-camera fusion for 3d object detection with transformers." Proceedings of the IEEE/CVF Conference on Computer Vision and Pattern Recognition. 2022.
Author Response
\documentclass[11pt]{article}
\usepackage{color, latexsym, times, verbatim, amsmath, graphics,epsfig,array, amssymb, colordvi}
\usepackage{algorithm}
\usepackage{color, xcolor}
% \usepackage{geometry}
% \geometry{top=1cm}
%\renewcommand{\baselinestretch}{2.0} %1.3
%\setlength{\oddsidemargin}{-0.7cm}
\setlength{\textwidth}{13.4625cm}
%\setlength{\evensidemargin}{-0.7cm}
\setlength{\columnsep}{0.9525cm}
\setlength{\textheight}{22.5425cm} \setlength{\hoffset}{0cm}
\setlength{\voffset}{0cm}\setlength{\topmargin}{-0.5cm}
%%%%%%%%%%%%%%%%%%%%%%%%%%%%%%%%%%%%%%%%%%%%%%%%%%%%%%%%%%%%%%%
\addtolength{\oddsidemargin}{-1.5cm}
\addtolength{\evensidemargin}{-2cm} \textwidth=16cm
\renewcommand{\baselinestretch}{1.2}
\begin{document}
\begin{center}
{\bf \LARGE {Responses to Reviewer 2}}
\end{center}
\large
\noindent\textbf{Manuscript ID:} remotesensing-2266845
\noindent\textbf{Title:} FusionRCNN: LiDAR-Camera Fusion for Two-stage 3D Object Detection
\noindent\textbf{Author:} Xinli Xu, Shaocong Dong, Tingfa Xu, Lihe Ding, Jie Wang, Peng Jiang, Liqiang Song, Jianan Li *
\\
\noindent We would like to thank you very much for your valuable comments and good suggestions that greatly helped to improve our manuscript. We have carefully considered your valuable comments and good suggestions. In the following, we are going to explain how your comments have been taken into full account in the revision. All modifications in the manuscript have been highlighted in \textcolor{yellow}{yellow}.\\
\noindent{\textbf{[Point1]}} {In general, this work extends prior multi-modal works to a two-stage manner.} It uses 3D proposals provided by LiDAR detectors and fuses camera features in the second stage with the popular attention mechanism. However, proposals generated by the first stage don’t use the multi-modal information, if the LiDAR points are too far or sparse, the proposal will not be detected as positive, so it’s perplexing that why not perform multi-modal fusion in the first stage. I think the consideration of this design choice should be clarified in this paper.
\noindent{\textbf{Response:}}
Thank you for your kind reminders. We clarified the consideration of this design choice in the revision, we revised the sentence as follows:\\
"According to our observation, fully trained one-stage point cloud detectors have high bounding box recall rates (IoU(Intersection over Union) \textgreater 0.3) even in cases where the faraway point clouds are sparse. However, the real challenge is that the lack of structural information in sparse point clouds leads to low confidence, poor localization, and incorrect classification of these proposal boxes, e.g., a car is misclassified as a bicycle with only several points within the proposal. Our novel two-stage fusion approach improves the precision of proposal boxes in a more accurate way. Although some well-designed one-stage fusion methods have achieved good performance, our method provides a new perspective for multi-modality fusion detection. We propose a two-stage plug-and-play refinement approach that can be attached as an additional enhancement module after any conventional detector without redesigning highly coupled and heavy network for each specific point cloud detector, bringing more flexibility."[Line64-76]\\
\noindent{\textbf{[Point2]}} All experiments in this article were conducted on the validation datasets, making it hard to compare with models in the official leaderboards. A better choice is to attach the test set results in the next edition.
\noindent{\textbf{Response:}}
Thank you for your kind suggestion. We attach the KITTI test set
results and give a careful analysis in the revision.\\
"For further validation, we report comparisons with previous methods on the KITTI test set in Table5. FusionRCNN achieves better performance than other competitive methods on the Moderate and Hard level, leading the state-of-the-art method CT3D by 0.36\%, 0.39\%, 0.33\% on Easy, Moderate and Hard level respectively. The experiments indicate that Our novel two-stage fusion framework is better at capturing structural and contextual information effectively compared with point-based two-stage methods"[Line242-348] \\
\noindent{\textbf{[Point3]}} In Table 8, it compares the model with the RoIAlign operation and with the RoIPooling operation, these results show a counterintuitive conclusion, however, this paper doesn’t give a detailed explanation. In general, RoIAlign performs better than RoIPooling, but table 8 shows quite the opposite. That’s an interesting point, please conduct more experiments and give some analysis.
\noindent{\textbf{Response:}}
Thank you for your kind reminders. We are also very interested in this point. It is difficult to explain why RoIPooling performs better than RoIAlign under certain conditions, but we think there may be a correlation with the expansion radio $k$. We further conduct experiments to investigate the influence of these two operations and expansion radio on our approach. The results are presented in Table9, and the revised analysis is as follows:
table9:\\
" FusionRCNN achieves better performance When expansion radio is set large (e.g., k=2.0), and there is little difference in performance between the two operations. This is attributed to the expansion of RoI that can effectively retain the foreground images corresponding to the proposals, ensuring that useful texture information can be utilized via LiDAR-camera fusion. When the expansion ratio is small, RoIAglin has better performance due to its more accurate RoI feature extraction ability, especially for far objects."[Line389-394]\\
Finally, thank you so much again for all your valuable comments and suggestions. We hope the manuscript after careful revisions has addressed all the concerns and meet your high standards. We welcome further constructive comments if any.
\end{document}

Round 2
Reviewer 2 Report
I am satisfied with the revised version.